# Two distinct DNA sequences recognized by transcription factors represent enthalpy and entropy optima

Ekaterina Morgunova[1]*, Yimeng Yin[1], Pratyush K Das[2], Arttu Jolma[1], Fangjie Zhu[1], Alexander Popov[3], You Xu[4], Lennart Nilsson[4], Jussi Taipale[1,2,5]*

[1]Department of Medical Biochemistry and Biophysics, Karolinska Institutet, Stockholm, Sweden; [2]Genome-Scale Biology Research Program, University of Helsinki, Helsinki, Finland; [3]European Synchrotron Radiation Facility, Grenoble, France; [4]Department of Bioscience and Nutrition, Karolinska Institutet, Huddinge, Sweden; [5]Department of Biochemistry, University of Cambridge, Cambridge, United Kingdom

**Abstract** Most transcription factors (TFs) can bind to a population of sequences closely related to a single optimal site. However, some TFs can bind to two distinct sequences that represent two local optima in the Gibbs free energy of binding ($\Delta G$). To determine the molecular mechanism behind this effect, we solved the structures of human HOXB13 and CDX2 bound to their two optimal DNA sequences, CAATAAA and TCGTAAA. Thermodynamic analyses by isothermal titration calorimetry revealed that both sites were bound with similar $\Delta G$. However, the interaction with the CAA sequence was driven by change in enthalpy ($\Delta H$), whereas the TCG site was bound with similar affinity due to smaller loss of entropy ($\Delta S$). This thermodynamic mechanism that leads to at least two local optima likely affects many macromolecular interactions, as $\Delta G$ depends on two partially independent variables $\Delta H$ and $\Delta S$ according to the central equation of thermodynamics, $\Delta G = \Delta H - T\Delta S$.

DOI: https://doi.org/10.7554/eLife.32963.001

*For correspondence:
ekaterina.morgunova@ki.se (EM);
ajt208@cam.ac.uk (JT)

**Competing interests:** The authors declare that no competing interests exist.

## Introduction

The binding of transcription factors (TFs) to their specific sites on genomic DNA is a key event regulating cellular processes. Analysis of structures of known TFs bound to DNA has revealed three different mechanisms of recognition of the specifically bound sequences: (1) the 'direct readout' mechanism involving the formation of specific hydrogen bonds and hydrophobic interactions between DNA bases and protein amino acids (*Aggarwal et al., 1988*; *Anderson et al., 1987*; *Wolberger et al., 1988*); (2) 'indirect readout' of the DNA shape and electrostatic potential (*Dror et al., 2014*; *Hizver et al., 2001*; *Joshi et al., 2007*; *Lavery, 2005*; *Rohs et al., 2005*) by protein contacts to the DNA backbone or the minor groove, and (3) water mediated interactions between bases and amino-acids (*Bastidas and Showalter, 2013*; *Garner and Rau, 1995*; *Ladbury et al., 1994*; *Morton and Ladbury, 1996*; *Patikoglou and Burley, 1997*; *Poon, 2012*; *Spolar and Record, 1994*). Each of these mechanisms contributes to binding specificity of most TFs, with their relative importance varying depending on the TF and the recognized sequence.

The modes of DNA recognition differ from each other also in their thermodynamic characteristics. For example, direct hydrogen bonds can contribute strongly to enthalpy of binding, whereas indirect hydrogen bonds mediated by water are weaker, due to the loss of entropy caused by immobilization of the bridging water molecule. In many cases, the contributions of the loss of entropy and the gain in enthalpy are similar in magnitude, leading to a phenomenon called 'enthalpy-entropy

**eLife digest** Genes are sections of DNA that carry the instructions needed to build other molecules including all the proteins that the cell needs to fulfill its role. The information in the DNA is stored as a code consisting of four chemical bases, often referred to simply as "A", "C", "G" and "T". The order or sequence of these bases determines the role of a protein. Many organisms – including humans – are built of many different types of cells that perform unique roles. Almost all cells carry the same genetic information, but proteins called transcription factors can regulate the activity of genes so that only a relevant subset of genes is switched on at a particular time.

Transcription factors glide along DNA and bind to short DNA sequences by attaching to the DNA bases directly or through bridges made up of water molecules. Two physical concepts known as enthalpy and entropy determine the strength of the connection. Enthalpy relates to how strong the chemical bonds that form between the transcription factors and the DNA bases are, compared to a situation where the transcription factor and DNA do not form a complex and bind to water molecules around them. Entropy measures the disorder of the system – the more disordered the solvent and protein-DNA complex are compared to solvent-containing free DNA and protein, the stronger the binding. A water molecule that bridges a DNA base with an amino-acid of a protein contributes to enthalpy, but results in loss of entropy, because the system becomes more ordered since the water molecule can no longer move freely.

Most transcription factors can only bind to DNA sequences that are very similar to each other, but some transcription factors can recognize several different kinds of sequences, and until now it was not clear how they could do this. Morgunova et al. studied four different human transcription factors that can each bind to two distinct DNA sequences. The results showed that the transcription factors bound to both DNA sequences with similar strength, but via different mechanisms. For one DNA sequence, an enthalpy-based mechanism essentially 'froze' the transcription factor to the DNA through rigid water bridges. The other DNA sequence was bound equally strongly but through moving water molecules, because this increased the entropy of the system. It is possible that these mechanisms could also apply to many other molecules that interact with each other through water-molecule bridges.

A better knowledge of the chemical bonds between transcription factors and DNA bases may in future help efforts to develop new treatments that depend on molecules being able to bind to other molecules. In addition, these findings may one day help scientists to predict how strongly two molecules will interact simply by knowing the structures of the molecules involved.

DOI: https://doi.org/10.7554/eLife.32963.002

compensation' (*Chodera and Mobley, 2013*; *Jen-Jacobson et al., 2000*; *Klebe, 2015*; *Patikoglou and Burley, 1997*). This leads to binding promiscuity, allowing a TF to bind to several different but closely related sequences with a biologically relevant affinity.

Many transcription factors appear to only recognize sequences closely related to a single optimal site. Their binding to DNA can be approximated by a position weight matrix (PWM) model, which describes a single optimal site, and assumes that individual substitutions affect binding independently of each other. Thus, the combined effect of multiple mutations is predictable from the individual effects. However, it is well established that several TFs can bind with high affinity to multiple different sequences, and populations of sequences that are closely related to these optimal sites (*Badis et al., 2009*; *Johnson et al., 2007*; *Jolma et al., 2013*; *Morris et al., 2011*; *Zhao and Stormo, 2011*; *Zuo et al., 2017*). In such cases, the effect of substitution mutations is not independent, and instead the mutations display strongly epistatic behavior (*Anderson et al., 2015*; *Jolma et al., 2013*; *Lehner, 2011*; *Zuo et al., 2017*), where the combined effect of two mutations can be less severe than what is predicted from the individual effects. Many cases of such multiple specificity can be explained by different spacing of homodimeric TFs (*Aggarwal et al., 1988*; *Párraga et al., 1998*), but in some cases a single monomeric TF appears to be able to bind to two distinct sequences with similar affinities.

The molecular mechanism behind the multiple specificity phenomenon has been understood at the structural level for some dimers (*Párraga et al., 1998*) but not for monomeric TFs. To elucidate

the mechanism, we performed structural analysis of two homeodomain proteins, the posterior home-odomain protein HOXB13, and the parahox protein CDX2, each bound to two distinct high-affinity sequences. The optimal sequences for HOXB13 are CCAATAAA and CTCGTAAA that differ from each other by the three underlined base pairs, whereas CDX2 binds with high affinity to similar two sequences that begin with a G instead of a C.

This analysis, together with thermodynamic measurements of HOXB13, CDX2 and two other transcription factors, BARHL2 and MYF5 that also display multiple specificity revealed that in each case, one of the optimal sequences is bound primarily due to an optimal enthalpic contribution, whereas the other is bound due to an optimum of entropy. This result is likely to be general to most macromolecular interactions, as they commonly involve interaction of the macromolecules with a network of interconnected water-molecules, whose formation involves a trade-off between enthalpy and entropy.

## Results and discussion

### Modeling the binding of many TFs requires more than one PWM model

Many TFs have been reported to display multiple specificity. These include many biologically important transcription factors such as the MYF family of basic helix-loop helix factors (*Yin et al., 2017*), the nuclear receptor HNF4A (*Badis et al., 2009*), and the homeodomain proteins BARHL2, CDX1 and HOXB13 (*Jolma et al., 2013*; *Jolma et al., 2015*; *Nitta et al., 2015*; *Zuo et al., 2017*). Analysis of enrichment of subsequences by MYF6, BARHL2, CDX1 and HOXB13 in SELEX reveals that a single PWM model cannot describe the binding affinity of these factors to DNA (*Figure 1A–D*). Each of these factors has more than one locally optimal sequence. All sequences between these optima have lower affinity and enrich less in SELEX than the optimal sequences. Therefore, more than one positionally independent position weight matrix (PWM) model is required for describing their affinity towards DNA (*Figure 1*).

Combinations of mutations affecting the optimal sites of these TFs display extremely strong epistatic effects. For example, the effect of mutating three first bases of the optimal HOXB13 motif TCGTAAAA is more than 400-fold smaller than what is expected from the individual single mutants (*Figure 1E,F*), and the generated CAATAAAA site binds to HOXB13 with almost the same affinity as the initial unmutated sequence.

### Structural analysis of HOXB13 and CDX2 bound to DNA$^{TCG}$ and DNA$^{CAA}$

To understand the molecular basis of the epistatic effect, we decided to solve the structure of HOXB13 and CDX2 bound to their two optimal sequences. These proteins are related, but diverged significantly in primary sequence, showing 43% identity at amino-acid level (*Figure 1—figure supplements 1* and *2*). For structural analysis, the DNA-binding domains (DBD) of HOXB13 (the 75 amino-acids from Asp-209 to Pro-283) and CDX2 (the residues Arg-154 – Gln-256) were expressed in *E.coli*, purified and crystallized bound to synthetic 19 or 18 bp double stranded DNA fragments containing the CTCGTAAA/GTCGTAAA (DNA$^{TCG}$) or CCAATAAA/GCAATAAA (DNA$^{CAA}$) motifs, respectively. These core sequences were obtained by PBM (*Berger et al., 2008*) and HT-SELEX (*Jolma et al., 2013*), and validated by ChIP-seq experiments (*Yin et al., 2017*), and represent the two distinct binding sites of HOXB13 and CDX2 (*Figure 2*). The structures were solved using molecular replacement at resolutions 3.2 and 2.2 Å for HOXB13, and 2.57 and 2.95 Å for CDX2, respectively.

All complexes displayed a high overall similarity to HOXB13 bound to methylated DNA (*Yin et al., 2017*), and to the previously known DNA-bound HOX protein structures (*Hovde et al., 2001*; *Joshi et al., 2007*; *LaRonde-LeBlanc and Wolberger, 2003*; *Passner et al., 1999*; *Piper et al., 1999*; *Zhang et al., 2011*) (*Figure 2*; *Figure 1—figure supplement 1*). Two parts of both HOXB13 and CDX2 DBDs interact with DNA: the recognition helix α3, which tightly packs into the major groove, and the N-terminal tail interacting with the minor groove (*Figure 2A,C*). The residue Gly-84 that is affected by a coding variant that is strongly implicated in prostate cancer was not included in our construct; two other residues mutated in single prostate cancer families

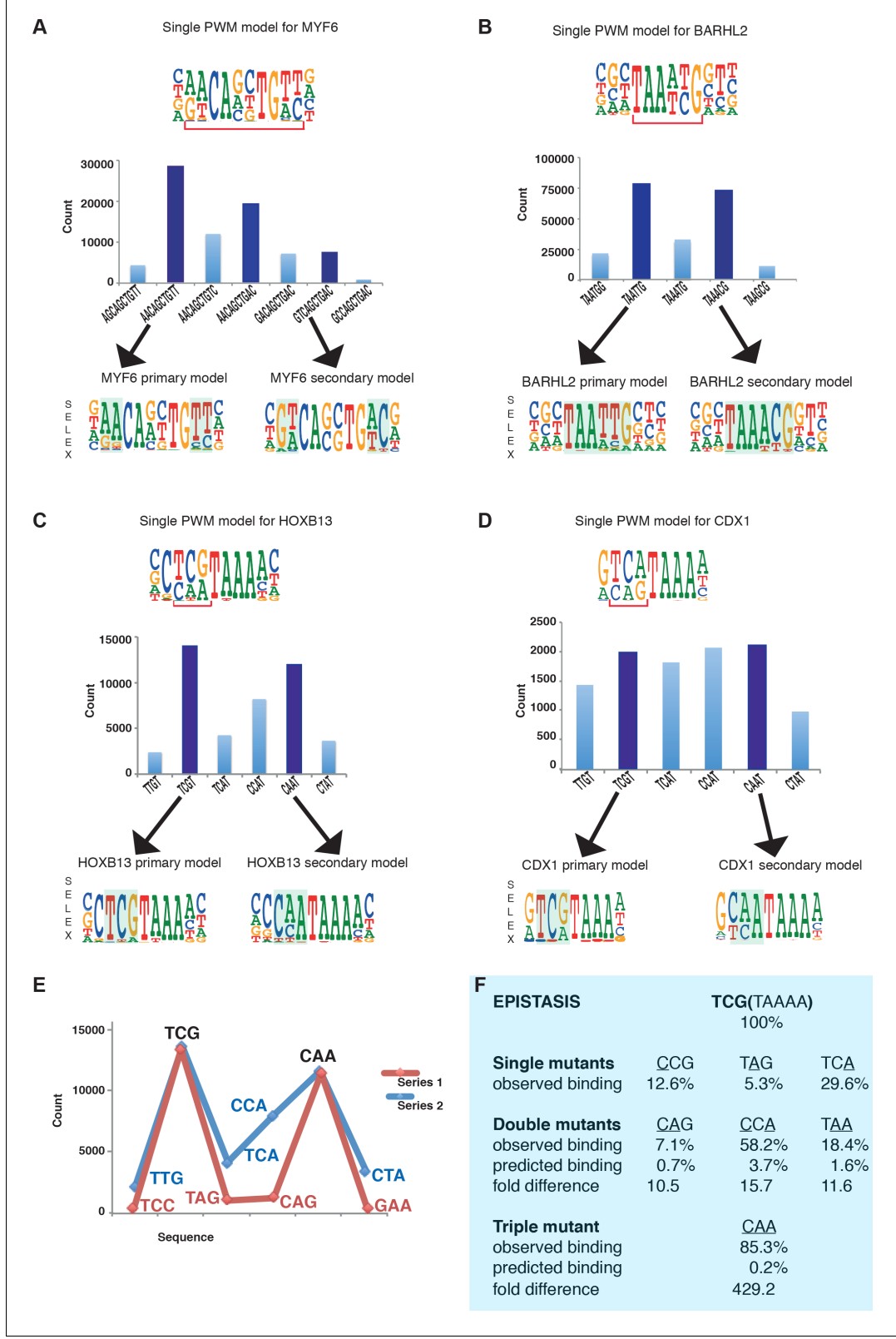

**Figure 1.** Multiple TFs prefer to bind to two optimal sequences. (**A**) MYF6 (this study); (**B**) BARHL2 (this study); (**C**) HOXB13 (*Yin et al., 2017*); (**D**) CDX1 (*Yin et al., 2017*). Note that single PWM models (top) fail to describe sequence specificity towards different sequences shown in the bar graphs (middle). For example, a single PWM model for HOXB13 (panel C, top) predicts near-equal affinities towards sequences TCG and TCA at the position of the bracket, and lower affinity towards CAA. Analysis of the counts of the subsequences (middle), instead, reveals that the TCA sequence is bound

*Figure 1 continued on next page*

*Figure 1 continued*

more weakly than the two most preferred sequences TCG and CAA. Counts for local maxima (dark blue) and related sequences that differ from the maxima by one or more base substitutions are also shown (light blue). The bars between the maxima represent sequences that can be obtained from both maximal sequences and have the highest count between the maxima. Bottom of each figure: Two distinct models that can represent the binding specificity of the TFs, the divergent bases are indicated by shading. For clarity, the PWM for the MYF6 optima that contains both AA and AC dinucleotide flanks (middle dark blue bar in A) is not shown. (E) Sequences representing the highest (blue line) and lowest (red line) affinity sequences between the two optimal HOXB13 sequences. y-axis: counts for 8-mer sequences containing the indicated trinucleotide followed by TAAA. (F) Epistasis in HOXB13-DNA binding. The effect of individual mutations (single mutants) to the optimal sequence TCGTAAAA (top) are relatively severe, with binding decreasing by more than 70% in all cases (observed binding). However, combinations of the mutations (double mutants) do not decrease HOXB13 binding in a multiplicative manner (compare predicted and observed binding). A multiplicative model predicts that combining all three substitutions would abolish binding, but instead the CAA site is bound more strongly than any other mutant (triple mutant).

DOI: https://doi.org/10.7554/eLife.32963.003

The following figure supplements are available for figure 1:

**Figure supplement 1.** The comparison of HOXB13 structure with HOXB1 and HOXA9.

DOI: https://doi.org/10.7554/eLife.32963.004

**Figure supplement 2.** HOXB13 prostate cancer mutation.

DOI: https://doi.org/10.7554/eLife.32963.005

(*Ewing et al., 2012*) were predicted to destabilize the protein (R229G), or its interaction with DNA (G216C) (*Figure 1—figure supplement 2*).

The core interactions between both HOXB13 and CDX2 DBDs and DNA are similar to those known from earlier structures (*Hovde et al., 2001*; *Joshi et al., 2007*; *LaRonde-LeBlanc and Wolberger, 2003*; *Passner et al., 1999*; *Piper et al., 1999*; *Yin et al., 2017*; *Zhang et al., 2011*). The TAAA sequence characteristic of the posterior homeodomains is recognized by a combination of a direct hydrogen bond to the $A_{10}$ base opposite of the T, and an insertion of the N-terminal basic amino-acids to the narrow minor groove induced by the stretch of four As. The overall protein structure in the four complexes is highly similar, showing only minor differences in the conformation of the N-termini, due to the replacement of basic Arg-217 of HOXB13 with the negatively charged Asp-187 in CDX2 (*Figure 2A, B, C, D*; *Figure 1—figure supplement 2B*). The most remarkable difference between the complexes is in the conformation of DNA of the HOXB13-DNA$^{TCG}$ complex at the position of the divergent bases (*Figure 2A,C*; *Figure 2—figure supplement 1*).

To quantitate the shape of the DNA in the protein binding region we determined the helicoidal parameters using the program Curves+ (*Lavery et al., 2009*), and found that the most prominent differences between the two complexes were in twist, shift, slide, X- and Y-displacement, minor groove width, and major groove depth at the positions of the divergent CAA and TCG sequences (*Figure 2—figure supplement 1B*). The DNA$^{TCG}$ backbone is bent towards the major groove, facilitating contact with Arg-258 of the recognition helix with the DNA backbone. The corresponding contact (Arg-228 to backbone) is also observed in both CDX2 structures. In contrast, the DNA$^{CAA}$ backbone is bent towards the minor groove, leading to a contact with N-terminal Arg-217 and Lys-218 (*Figure 2B*; *Figure 3*). Instead of contacting the DNA backbone, Arg-258 assumes an alternative conformation in which it turns inside of the major groove, forming a water-mediated contact with Gln-265. The Gln-265, in turn, recognizes $C_{6'}$ via a direct hydrogen bond. In addition, the CAA sequence is recognized by a hydrophobic interaction between Ile-262 and the $T_{11}$ methyl group. In CDX2 complexes the DNA bend in CDX2-DNA$^{TCG}$ is slightly smaller due to the replacement of Thr-261 with Lys-231 which does not allow the alternative conformation of Arg-228. The other contacts in the CDX2-DNA complexes are very similar to those listed for HOXB13-DNAs.

In order to understand the role of individual residues in binding of specific DNA we created 48 different single and combined mutations in DBD of HOXB13. This set of experiments was carried out to mechanistically understand the origin of the dual specificity, and the molecular mechanisms that explain different specificities between the posterior HOX proteins. The resulting data are presented in *Figure 2E* and *Figure 2—figure supplement 2*. The replacement of Thr-261 either as a single mutation or in combination with any other amino-acids resulted in changing of HOXB13 specificity from Ctcg/Ccaa towards the sequence recognized by CDX2 (Gtcg/Gcaa; *Figure 2E*, left panel). No substitutions were identified that would lead to a specific and complete loss of binding to either the TCG or CAA sequence. However, some weaker effects were detected; several mutations affecting

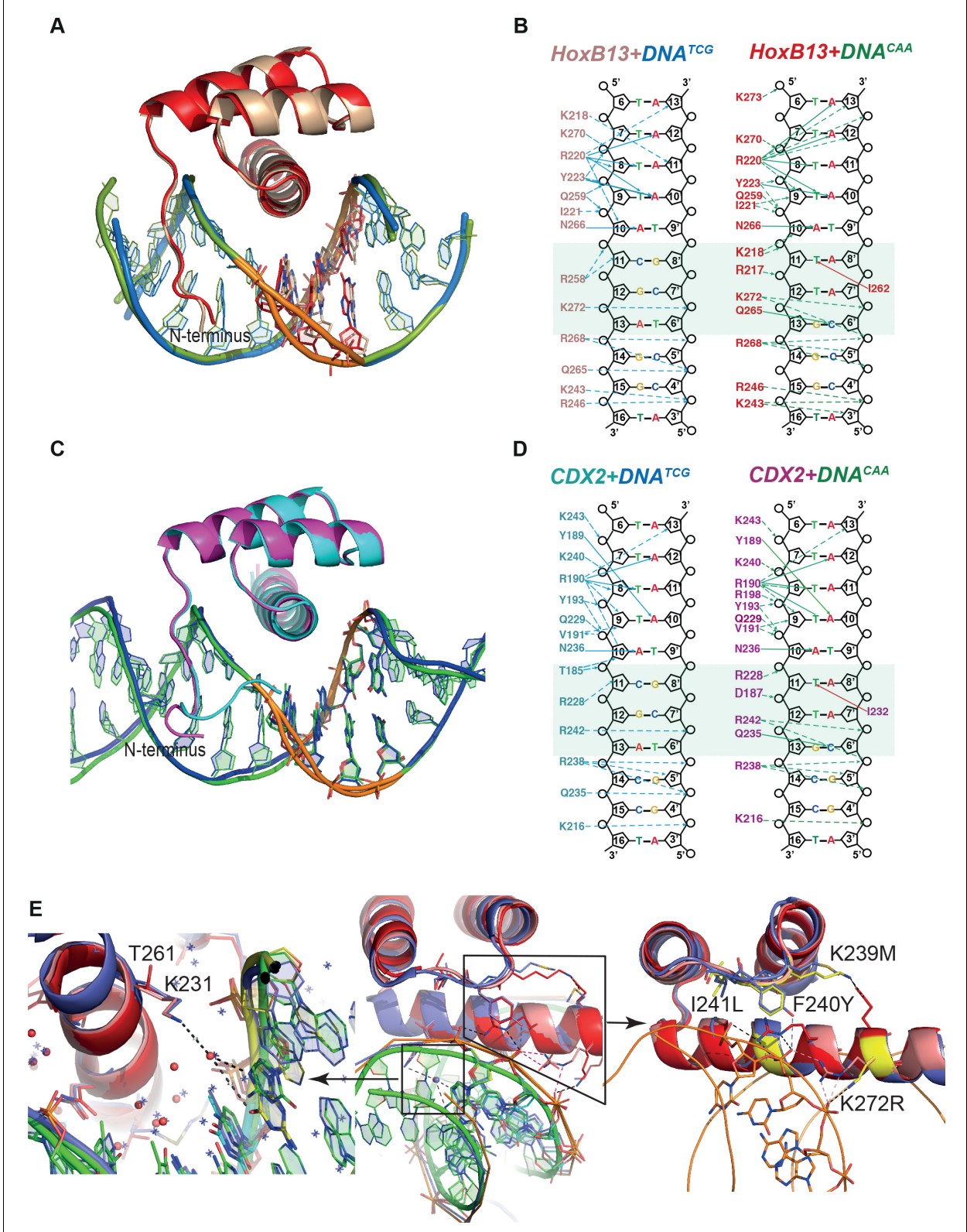

**Figure 2.** Comparison of Protein-DNA complexes. (**A**) The view of superposition of HOXB13 (wheat) bound to DNA$^{TCG}$ and HOXB13 (red) bound to DNA$^{CAA}$ (rmsd = 0.813 Å on 57 residues). The respective DNAs are in blue and green. The dissimilar base pairs are presented as ball-and-stick models and colored as the proteins, DNA$^{TCG}$ is wheat and DNA$^{CAA}$ is red. Note the different bending of the DNA backbone at these positions (orange). (**B**) Schematic representation of interactions formed between HOXB13 DBD and the two different DNAs: left panel shows the interactions between

*Figure 2 continued on next page*

*Figure 2 continued*

HOXB13 and the primary binding site (DNA$^{TCG}$) and right panel represents the interactions of HOXB13 with the secondary site (DNA$^{CAA}$), respectively. Dashed lines represent interaction with backbone phosphates and deoxyribose and solid lines interactions with the bases. The protein residues belonging to the HOXB13-DNA$^{TCG}$ and HOXB13-DNA$^{CAA}$ structures are colored wheat and red, respectively. The divergent parts of the DNA sequences are highlighted by a light green box. Note that the TCG site lacks direct contacts to the DNA bases, whereas the CAA site is recognized by direct contacts by Gln-265 and Ile-262. Most other contacts are similar in both structures. The four As of the TAAAA sequence are recognized by the N-terminal amino-acids interacting with the DNA backbone via the minor groove, whereas the T is recognized by a bidentate interaction formed between its complementary adenine $A_{10}$ and the side chain of asparagine Asn-266. Two hydrogen bonds are formed between nitrogen atoms $N^6$ and $N^7$ from adenine base and oxygen and nitrogen atoms of the Asn-266 side chain. This adenine-specific asparagine is totally conserved in the HOX family. (C) Superposition of CDX2 (cyan) bound to DNA$^{TCG}$ and CDX2 (magenta) bound to DNA$^{CAA}$ (rmsd = 0.270 Å on 64 residues). The respective DNAs are in blue and green. The dissimilar base pairs are presented as ball-and-stick models and colored as the proteins, DNA$^{TCG}$ is green and DNA$^{CAA}$ is blue. Note the different bending of the DNA backbone at these positions (orange). (D) Schematic representation of interactions formed between CDX2 DBD and the two different DNAs. (E) Structural interpretation of mutations that change the specificity of HOXB13: the mutations changing Ccaa/Ctcg to Gcaa/Gtcg are shown in a small box and, as a close view, on the left panel, and mutations, which switch the preferences of HOXB13 from CTCG to CCAA, are shown in big box and, as a close view, on the right panel. The mutations are presented in structural alignment of HOXB13 (red), HOXA9 (blue, PDB entry 1PUF) and CDX2 (pink) bound to DNA. Note the unique mutation of Lys (small box), which is conserved in all known HOXes, to Thr in HOXB13 allows HOXB13 to accept any base pair in the position before TCG/CAA. The left panel is representing the close view to the interactions formed by Lys in HOXA9 and CDX2. Long aliphatic chain of Lys increases the hydrophobicity of this part of protein-DNA interface, pushing out the water molecules. Dashed line indicates water-mediated interaction between the ε-Amino group of Lys and the N7 and O6 of the guanine base at the Gtcg sequence. The right panel is representing the close view of triple mutation in the loop connecting helix 1 and helix 2: Lys-239/Met, Phe-240/Tyr and Ile-241/Leu; and single mutation of Lys-272/Arg. Those mutations are expected to change the hydrogen bond network between the protein and DNA and lead to a preference towards the more rigid, more B-shaped DNA$^{CAA}$.

DOI: https://doi.org/10.7554/eLife.32963.006

The following figure supplements are available for figure 2:

**Figure supplement 1.** Paiwise comparison of two DNA molecules.
DOI: https://doi.org/10.7554/eLife.32963.007

**Figure supplement 2.** HOXB13 - HOXes/CDX mutations.
DOI: https://doi.org/10.7554/eLife.32963.008

backbone contacts between HOXB13 amino-acids and DNA 5' of the divergent trinucleotide moderately increased the relative affinity towards the CAA sequence (*Figure 2E*, right panel).

Analysis of the mutation data together with the structures revealed that amino-acids involved in the protein-DNA interface formation cannot fully explain the specificity preferences of HOXB13 and CDX2. The lack of direct interactions between protein and DNA in this region instead suggests that the specificity would be conferred in part by bridging water-molecules located at the protein-DNA interface.

## Role of water molecules in the protein-DNA interface

The main difference between the complexes with DNA$^{CAA}$ and DNA$^{TCG}$ is revealed by analysis of the bridging water molecules. The HOXB13-DNA$^{CAA}$ structure (2.2 Å) contains chains of water molecules that interact with both HOXB13 amino-acids and each of the DNA bases in the CAA sequence (*Figure 4A–C*). In contrast, no water molecules are visible in the HOXB13-DNA$^{TCG}$ structure, despite the 3.2 Å resolution that should allow identification of strongly bound water molecules. Consistently, only few water molecules were found in the CDX2-DNA$^{TCG}$ complex. A relatively large solvent channel (6.4 Å in smallest diameter) exists between the α3 helix of HOXB13 and DNA$^{TCG}$ (*Figure 4D*) while this space is occupied by well-defined water-net including Arg-258 in HOXB13-DNA$^{CAA}$ structure (*Figure 4—figure supplement 1A*). The electron density in this region of HOXB13-DNA$^{TCG}$ structure is low (σ < 0.5), similar to that found in the surrounding solvent, indicating that the water-molecules in this region are highly mobile. Thus, the optimal binding of HOXB13 to the CAA sequence can be rationalized by the visible interactions that contribute to the enthalpy of binding (ΔH). In contrast, no such interactions can be identified that could explain the preference of HOXB13 to the TCG trinucleotide. The absence of ordered solvent molecules, and the lower resolution of the HOXB13-DNA$^{TCG}$ structure is consistent with the possibility that the TCG sequence is preferred because it represents a relatively disordered, high entropy state. In complex of CDX2-DNA$^{TCG}$ with high resolution (2.57 Å) the water molecules were well visible but they did not form the corresponding water-chains (*Figure 4E,F*) supporting the idea of entropically driven binding. The channel is not

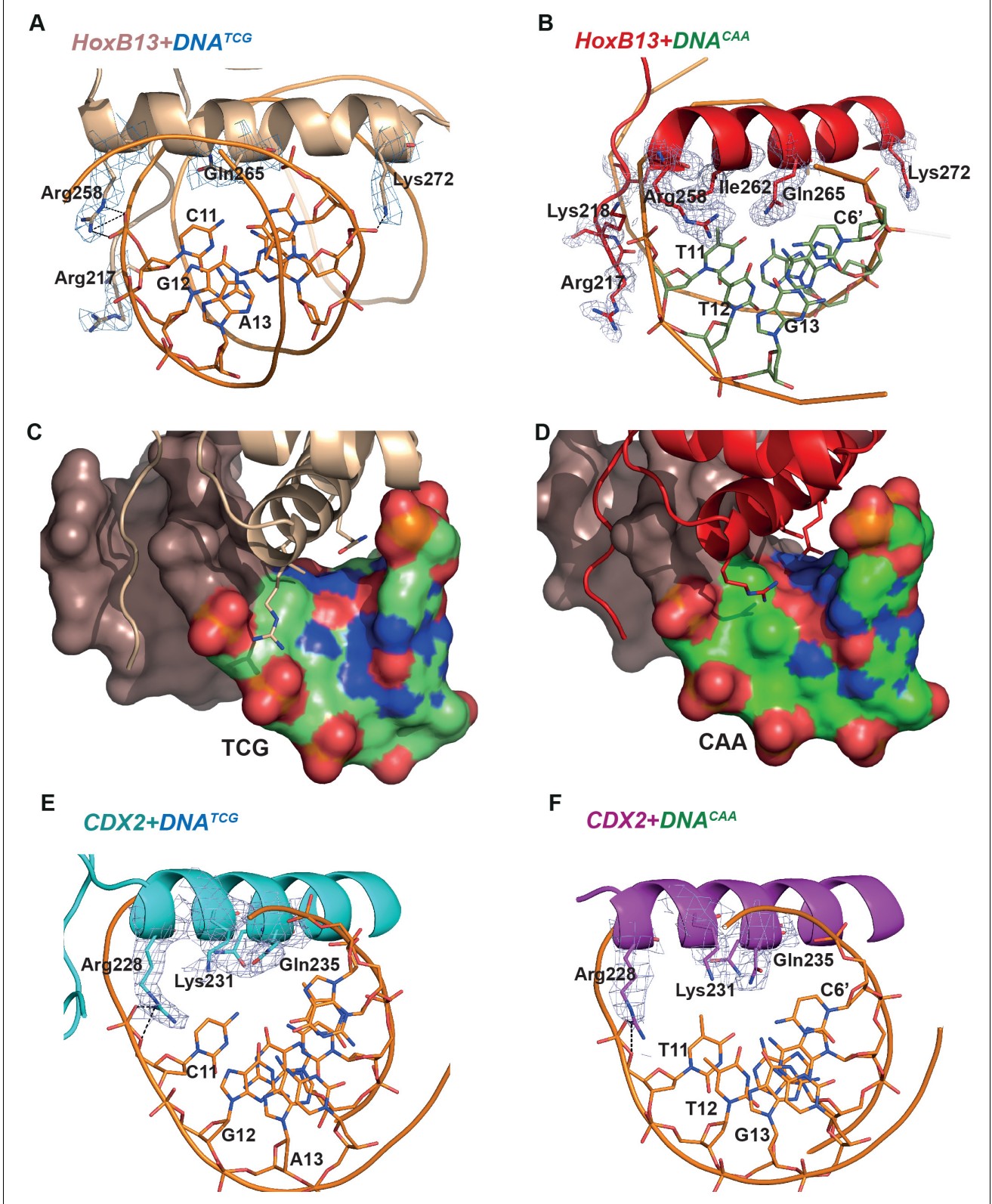

**Figure 3.** Close view of the protein-DNA interactions. (A) HOXB13-DNA[TCG] and (B) HOXB13-DNA[CAA] complexes. The 2mFo-Fc maps contoured with 1.5σ are shown around the key residues. The residues and base pairs involved in protein to DNA contacts are also labeled. (C, D) Surface representation of the major groove in HOXB13-DNA[TCG] and HOXB13-DNA[CAA] complexes, respectively. The divergent bases are colored to indicate electrostatic charges of the atoms: neutral carbon atoms are green, oxygen atoms (negative) are red and nitrogen atoms (positive) are blue. Note the

*Figure 3 continued*

larger solvent-accessible space between amino-acids and bases in the TCG structure (**C**) and the difference in distribution of the positively and negatively charged spots on the surface that can contribute to differences in distribution of water molecules on the surface. (**E**) CDX2-DNA$^{TCG}$ and (**F**) CDX2-DNA$^{CAA}$ complexes. The 2mFo-Fc maps contoured with 1.5σ are shown around the key residues. The residues and base pairs involved in protein to DNA contacts are also labeled.

DOI: https://doi.org/10.7554/eLife.32963.009

as clearly visible from one side of the CDX2-DNA$^{TCG}$ complex because of the presence of Lys-231 instead of Thr-261. The long side chain of Lys-231 does not allow Arg-228 (corresponded to Arg-258 in HOXB13) to change its conformation easily. Together, Arg-228 and Lys-231 close the entry of the channel from this side but the other end of the channel is open and well visible in CDX2-DNA$^{TCG}$ (*Figure 4—figure supplement 1B,C*).

## Thermodynamic features of the protein-DNA interactions

We next performed molecular dynamics simulations and free energy perturbation calculations to probe the behavior of water molecules in the protein-DNA interface for the two optimal sequences for HOXB13. The relative free energy (*Hansson et al., 1998*) estimates for the affinities of HOXB13 for the two DNA sequences obtained from the simulations indicate that both sequences are bound with similar affinities ($\Delta\Delta G$ = - 0.1 kcal/mol). Analysis of the mobility of water molecules at the protein-DNA interface revealed that, while there is a similar number of water molecules in both systems, the waters at the HOXB13-DNA$^{TCG}$ interface are more mobile (*Figure 5—figure supplement 1*), consistent with a model where this complex has higher entropy than the HOXB13-DNA$^{CAA}$ complex.

To more directly test if the two states are driven by enthalpy and entropy, we measured these thermodynamic parameters using isothermal titration calorimetry (ITC). ITC directly measures the heat of binding ($\Delta H$) and $K_d$ of the binding reaction. Conversion of the $K_d$ to $\Delta G$ then allows the inference of the entropy of binding ($\Delta S$) from the data. The measured thermodynamic parameters for the TCG site were very similar to those we reported previously (*Yin et al., 2017*). Comparison of the parameters for the TCG and CAA sites revealed that consistent with SELEX (*Jolma et al., 2013*) and molecular modeling data, the $\Delta G$ values for both sequences were similar. However, as predicted, the CAA site displayed much higher change in enthalpy, and larger loss of entropy compared to those of the TCG site (*Figure 5A,B*, *Supplementary file 1*). These results indicate that HOXB13 binding to one optimal site, CAA, is driven by enthalpy, whereas strong binding to the other, TCG, is due to a lower loss of entropy.

To test if the identified mechanism is general to other cases of multiple specificity, we used ITC to determine the thermodynamic parameters for CDX2 and two other TFs, the MYF family TF MYF5 and the homeodomain protein BARHL2, both of which can optimally bind to two distinct sequence populations. Analysis of the data confirmed that in both cases, the $\Delta G$ values for the two optimally bound sequences were similar, whereas the relative contributions of entropy and enthalpy to the binding were strikingly different (*Figure 5C–H*). These results suggest that the ability of some TFs to bind to two distinct sequences with high affinity can be caused by the presence of both an enthalpic and an entropic optima.

## Conclusions

In drug development, multiple optimal compounds can often be found that bind to a particular target molecule (*Klebe, 2015*). However, biological macromolecules are composed of a small set of relatively large monomers, and thus populate the shape-space more sparsely than synthetic small molecules, which can be modified at the level of single atoms. Therefore, the finding that TFs can bind to two distinct DNA sequences with equal affinity was unexpected (*Badis et al., 2009*; *Jolma et al., 2013*) and has been controversial in the field. Our initial hypothesis was that the two optimal states could be due to an ability of the TF to adopt two distinct conformational states, or due to a similarity of the shapes of the two distinct DNA sequences (*Kalodimos et al., 2002*; *Nakagawa et al., 2013*; *Párraga et al., 1998*). To address these hypotheses, we solved the structure of HOXB13, a central transcription factor involved in both development (*Economides and Capecchi, 2003*; *Krumlauf, 1994*; *Nolte et al., 2015*) and tumorigenesis (*Ewing et al., 2012*;

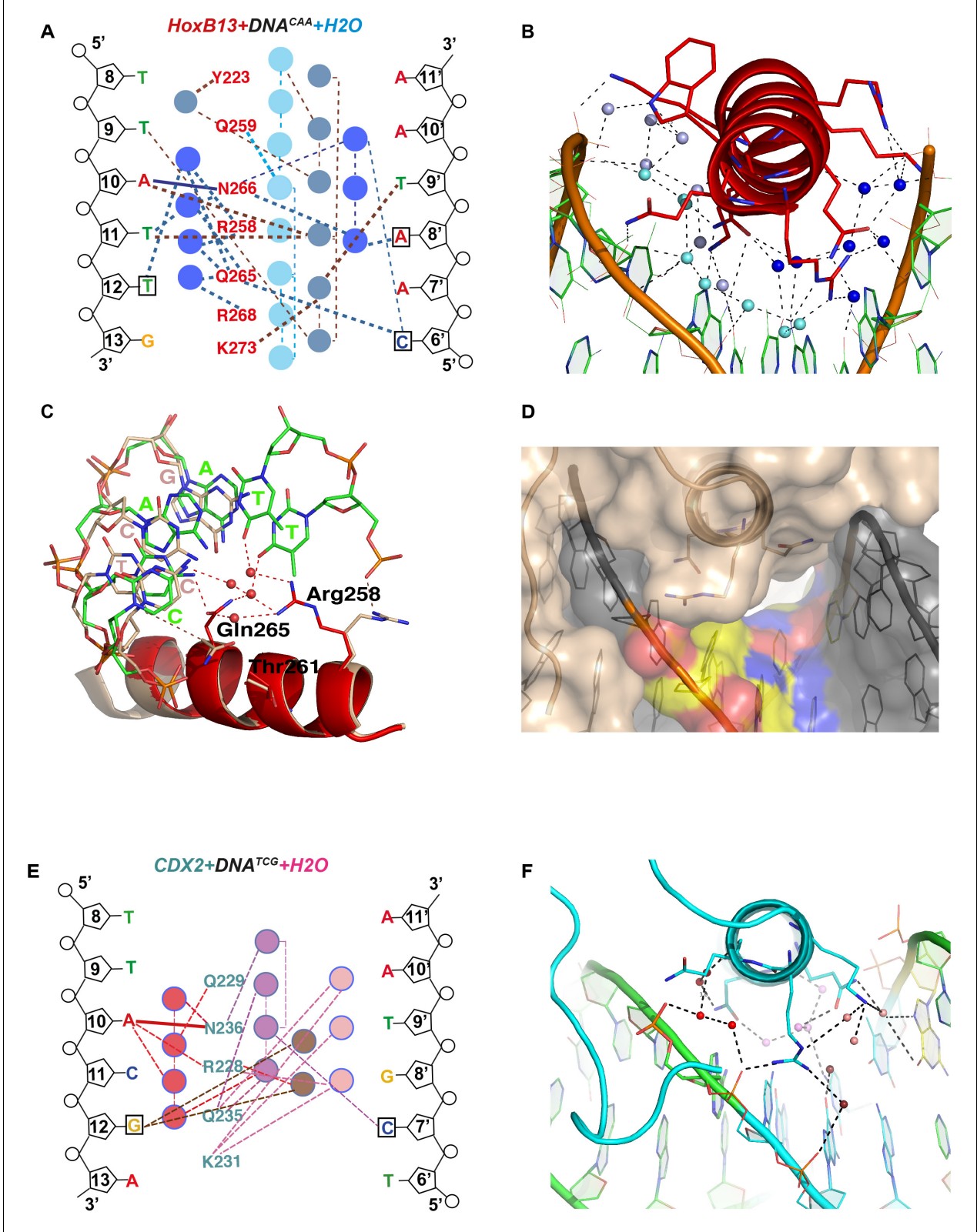

**Figure 4.** Close view of the role of water molecules in HOXB13-DNA interaction. (**A**) Schematic representation of water-mediated interactions between amino-acids (red typeface) of HOXB13 and DNA bases in the HOXB13-DNA$^{CAA}$ structure. Different water chains are indicated with different shades of blue. Thick dashed lines represent interactions formed between water molecules and bases or amino acids; thin dashed lines represent contacts formed between water molecules, and solid blue line indicates the direct interaction between A$_{10}$ and Asn-266. Note that all of the base positions in

*Figure 4 continued on next page*

*Figure 4 continued*

the CAA sequence (boxes) are recognized via direct or water-mediated hydrogen bonds. (**B**) Structural representation of the network of interactions schematically presented in (**A**). Note the three water chains colored by slightly varied blue color. The amino acids and bases involved in interactions are presented as stick models. (**C**) Close view to the different conformations of amino-acids observed in HOXB13-DNA$^{TCG}$ and HOXB13-DNA$^{CAA}$ structures. Note that the conformations of the key amino-acids Gln-265 and Arg-258 that interact with the water network in HOXB13-DNA$^{CAA}$ (amino-acids in red, DNA carbons in green) are not suitable for interacting with the network in HOXB13-DNA$^{TCG}$ (amino-acids and DNA carbons in wheat). (**D**) Surface representation of protein-DNA interface of HOXB13-DNA$^{TCG}$ complex. Relatively large channel between the protein and DNA that goes through the protein-DNA interface (white) lend support to the presence of mobile water molecules in this region. TCG-bases are colored by atoms: carbon atoms are yellow; oxygen atoms are red and nitrogen atoms are blue. (**E**) Schematic representation of water-mediated interactions between amino-acids (cyan typeface) of CDX2 and DNA bases in the CDX2-DNA$^{TCG}$ structure. Different water chains are indicated with different shades of red. Thick dashed lines represent interactions formed between water molecules and bases or amino acids; thin dashed lines represent contacts formed between water molecules, and solid red line indicates the direct interaction between $A_{10}$ and Asn-236. Note that only the position of the GC pair is recognized (boxes) via water-mediated hydrogen bonds. (**F**) Structural representation of the network of interactions schematically presented in (**E**). Note the three water chains colored by varied red-pink color. The amino acids and bases involved in interactions are presented as stick models.

DOI: https://doi.org/10.7554/eLife.32963.010

The following figure supplement is available for figure 4:

**Figure supplement 1.** Surface representation of protein-DNA interface of HOXB13:DNA$^{CAA}$ complex (A); CDX2:DNA$^{CAA}$ (B) and CDX2:DNA$^{TCG}$ (C).

DOI: https://doi.org/10.7554/eLife.32963.011

*Huang and Cai, 2014*; *Pomerantz et al., 2015*) bound to its two optimal DNA sequences. Surprisingly, the conformational differences between the HOXB13 proteins in the two structures were minor. Similarly, the two CDX2-DNA structures also displayed very similar overall conformations. In addition, the shape and charge-distribution of the optimally bound DNA sequences were not similar to each other. Thus, the structural analysis failed to support either the dual protein conformation or the DNA shape similarity models. Instead, thermodynamic analyses of HOXB13, CDX2, BARHL2 and MYF5 revealed that the two optimal states were bound because of their distinct effects on enthalpy and entropy, principally caused by differential stability of the water network at the protein-DNA interface. We also failed to find mutations in HOXB13 that would specifically abolish binding to one of the DNA sites. This finding is consistent with the thermodynamic model presented, as both TCG and CAA are bound by the same conformation of the HOXB13 protein, using the same contacting amino-acids (even when the contacts can occur via water or be direct).

Our findings explain the mechanism behind the observed genetic epistasis at some TF binding sites. In addition, our findings are relevant to evolution of TF binding sites and macromolecular interactions. The ability of one molecular shape (HOXB13) to bind to two distinct molecular shapes (CAA and TCG) with similar affinity allows other mechanisms to selectively modulate target sites in the genome. For example, one of the HOXB13 optima, TCG, is affected by DNA methylation (*Yin et al., 2017*; *Zuo et al., 2017*), and methylation of this site further increases HOXB13 affinity. The CAA site does not contain a CpG dinucleotide, and thus cannot be inheritably methylated. Thus, the ability of HOXB13 to bind to two distinct sites with similar affinity allows evolution of two types of regulatory sequences, those that are directly and positively affected by CpG methylation, and those that are not.

The mechanism by which TFs bind to two optimal DNA sequences is fundamental, and applies to all macromolecular interactions. In principle, enthalpy and entropy of binding vary partially independently as a function of the shape and charge distribution of the interacting molecules. Thus, different sequences are likely to be optimal with respect to enthalpy and entropy, in such a way that one optimal sequence is close to the optimal enthalpy state, and another is close to the optimal entropy state (*Figure 6*). We are not aware of work that has previously described such a situation. The observed effect that commonly results in at least two distinct ΔG minima may have been missed before because it is generally strongest when there are solvent molecules at the interacting surface. Interacting through solvent molecules can increase enthalpy of binding, but also causes a large loss of entropy due to fixing of the solvent molecule(s). However, in macromolecular interactions that are driven by direct contact between residues, entropy often has lower impact on binding than enthalpy, and thus one of the optima is at a higher ΔG than the other. Another reason for overlooking this mechanism could have been the fact that multiple local optima can also exist via other mechanisms (see for example [*Klebe, 2015*]), and measurements that allow inference of entropy and enthalpy

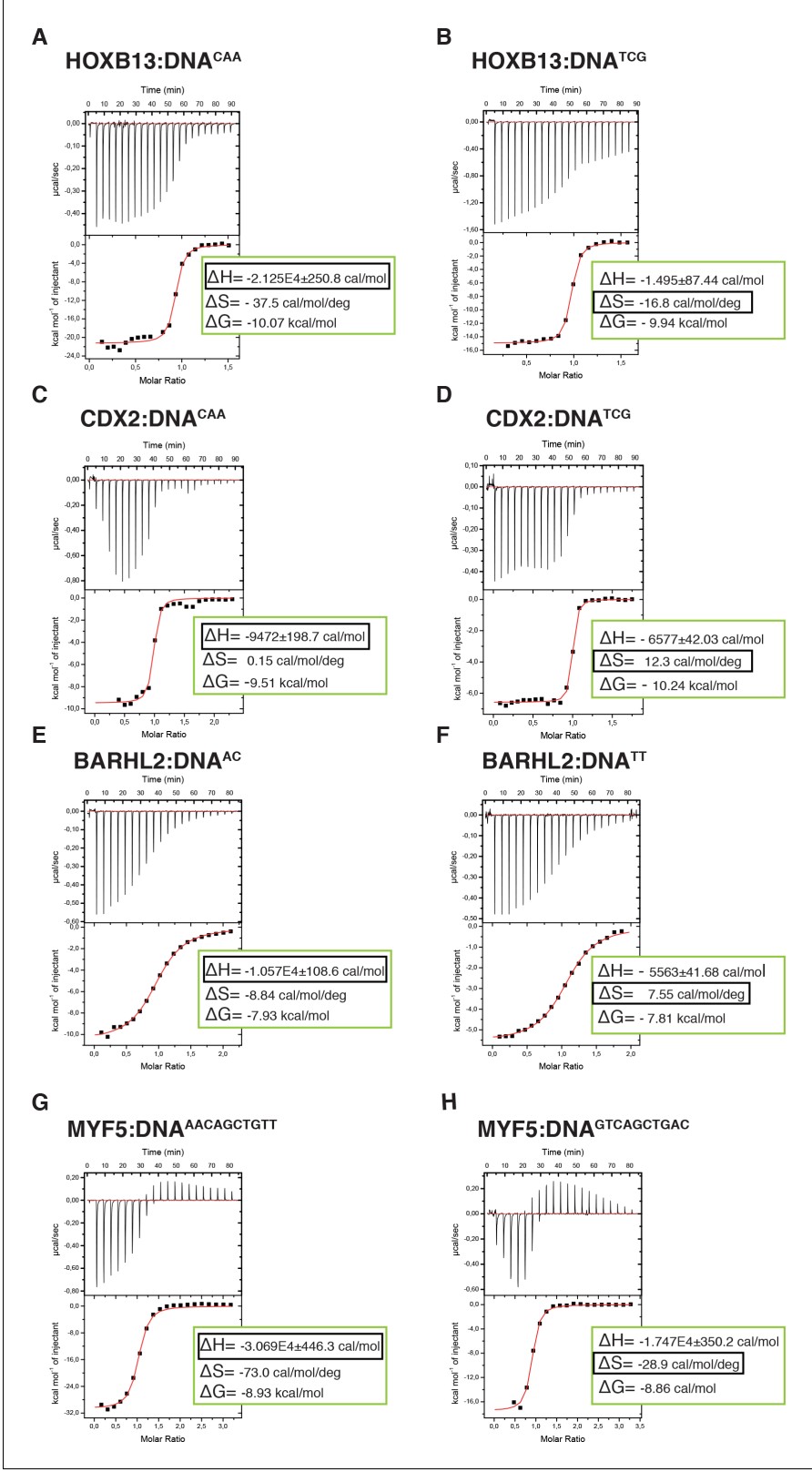

**Figure 5.** Calorimetric titration data reveals that two optimal DNA sequences recognized by HOXB13 (**A**, **B**), CDX2 (**C**, **D**), BARHL2 (**E**, **F**) and MYF5 (**G**, **H**) represent enthalpy and entropy optima. The optimal sequences with higher enthalpic contribution to binding are presented on the left side (**A**, **C**, **E**, **G**) and the reactions with higher entropic contribution are presented on the right side (**B**, **D**, **F**, **H**). Note that for each protein both DNAs are bound with

*Figure 5 continued on next page*

*Figure 5 continued*

similar ΔG. The top panels of the ITC figures represent raw data; the bottom panels show the integrated heat of the binding reaction. The red line represents the best fit to the data, according to the model that assumes a single set of identical sites. The determined changes of enthalpy and calculated losses of entropy are shown on the bottom panel. The changes of Gibbs free energy, ΔG=ΔH-TΔS, are also calculated and presented on the bottom panel of each isotherm.

DOI: https://doi.org/10.7554/eLife.32963.012

The following figure supplement is available for figure 5:

**Figure supplement 1.** Distribution of water-bridge lifetimes in HOXB13:DNA complexes.

DOI: https://doi.org/10.7554/eLife.32963.013

separately are not commonly performed in studies of macromolecular interactions. In addition, simple additive binding models such as position weight matrices (PWMs) can hide the effect, as they can only describe a single optimal state.

The cases we studied here represent some of the strongest deviations from the PWM model, and also present two optima of very similar ΔG that are located relatively far from each other in sequence space. Because we selected for this study cases with two almost equally strong binding sequences, the effect is likely to be stronger here than in most other interactions. Thus, the relative importance of the phenomenon across other types of interactions needs to be evaluated experimentally. The phenomenon we observed exists due to the fact that entropy and enthalpy of binding vary as a function of the shape of the interacting macromolecules. Both functions will have optima, commonly at different exact positions. The binding free energy will also vary as a function of the shape, but is likely to have more optima than its constituent functions because they are partially independent of each other. Thus, it is likely that many other biologically relevant examples of this effect will be identified. Many of these will represent cases where the sequences representing entropic and enthalpic optima are more closely related to each other than in the extreme cases studied here. This would manifest as a 'flat bottom' in the affinity landscape, where many sequences would bind with similar affinity. In addition, it is likely that in most cases one of the local optima is located at lower affinity than the other. This would manifest as a minor peak or a shoulder in the affinity landscape away from the optimal sequence. In each case, the measured affinities would deviate from those predicted from a single PWM model.

Our results and the underlying theory suggest that the ability of TFs to bind to distinct sequences could thus be widespread, and that the importance of the optimal states in determining TF-DNA binding preferences should be reinvestigated. Moreover, models for TF binding that are used to identify TF sites should also be adjusted to include features that allow two or more optima. In a broader sense, our results are potentially relevant to all macromolecular interactions, particularly in the presence of a polar solvent such as water that can contribute to bridging interactions, whose contributions to the enthalpy and entropy of binding are in the same order of magnitude. Therefore, in addition to explaining the observed epistasis in protein-DNA interactions, the presence of two optima is likely to also explain the molecular mechanisms behind other types of genetic epistasis.

## Materials and methods

### Protein expression, purification and crystallization

Expression and purification of the DNA-binding domain fragment of human HOXB13 (residues 209–283) as well as CDX2 (residues 184–256) were performed as described in Refs. (*Savitsky et al., 2010*) and (*Yin et al., 2017*). The DNA fragments used in crystallization were obtained as single strand oligos (Eurofins), and annealed in 20 mM HEPES (pH 7.5) containing 150 mM NaCl and 0.5 mM *Tris* (2-carboxyethyl) phosphine (TCEP) and 5% glycerol. For each complex, the purified and concentrated protein was first mixed with a solution of annealed DNA duplex at a molar ratio 1:1.2 and after one hour on ice subjected to the crystallization trials. The crystallization conditions for all complexes were optimized using an in house developed crystal screening kit of different PEGs. Complexes were crystallized in sitting drops by vapor diffusion technique from solution containing 50 mM Tris buffer (pH 8.0), 100 mM $MgCl_2$, 150 mM KCl, 8% of PEG (400) and different concentrations

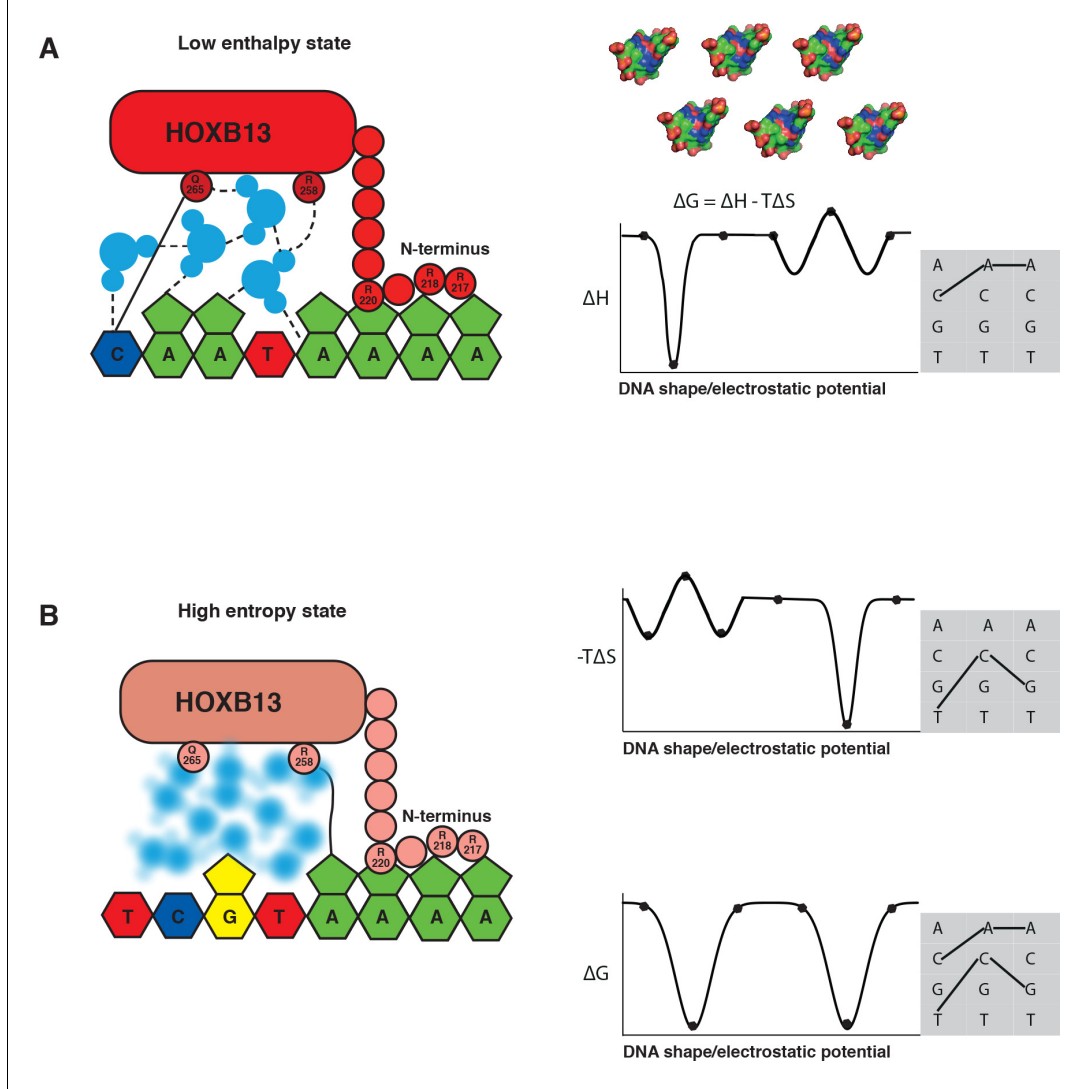

**Figure 6.** The two optimal sites bound by HOXB13 represent enthalpy and entropy-driven optima. (A–B) Schematic cartoon illustrations of the binding mechanism driven by the low enthalpy (A) and by high entropy (B) are presented in the left panels. The DNA bases are presented as pyrimidine and purine rings, protein is represented as ellipsoid, N-terminus is shown bound to the minor groove created by A-stretch, and water molecules are shown schematically and colored blue. The dashed lines represent hydrogen bonds observed in the low enthalpy state; the solid line represents direct interactions between amino acids and bases. The blurred water molecules indicate the high entropy state. Hydrogen bonds that are common to both complexes are omitted for clarity. Graphs on the right show schematic illustration of the variance of enthalpy ($\Delta H$, top), entropy ($-T\Delta S$, middle) and Gibbs free energy ($\Delta G$) (bottom) as a function of an idealized one-dimensional continuous variable representing the high-dimensional variables of shape, electrostatic charge and vibration of DNA that vary as a function of the DNA sequence. As DNA is composed of only four bases, only discrete positions along this axis are possible (indicated by dots). Example models of shape and charge distribution of different DNA sequences (from *Figure 1C*) are shown as surface representation above the scheme. The surfaces are colored according to the charge distribution: positively charged atoms are in blue, negatively charged are in red and neutral atoms are in green. Note that enthalpy and entropy are partially negatively correlated, leading to binding promiscuity (wider optima in $\Delta G$ compared to $\Delta H$ and $\Delta S$). The remaining uncorrelated component leads to the presence of two optima for $\Delta G$ (bottom). Shaded boxes on the right show simplified dinucleotide binding models that illustrate how this leads to two distinct locally optimal sequences. Note that the values are for illustration purposes only and the 'bumps' illustrate local entropy-enthalpy compensation that broadens the peaks of $\Delta G$.

DOI: https://doi.org/10.7554/eLife.32963.014

of various PEGs. PEG (3350) was used at 14% for HOXB13-DNA$^{TCG}$ and 21–27% (w/v) of polyethylene glycol monomethyl ether (PEGmme (5000)) was used in crystallizations of HOXB13 with DNA$^{CAA}$ and CDX2 with both DNAs. The data sets were collected at ESRF from a single crystal on beam-lines ID29 (HOXB13-DNA$^{TCG}$) and ID23-1 (HOXB13-DNA$^{CAA}$, and both CDX2 complexes), at 100 K using the reservoir solution as cryo-protectant. The data collection strategy was optimized with the program BEST (*Bourenkov and Popov, 2006*). Data were integrated with the program XDS (*Kabsch, 2010*) and scaled with SCALA (*Murshudov et al., 2011*; *Winn et al., 2011*). Statistics of data collection are presented in *Table 1*.

## Structure determination and refinement

All structures were solved by molecular replacement using program Phaser (*McCoy et al., 2007*) as implemented in Phenix (*Adams et al., 2010*) and CCP4 (*Winn et al., 2011*) with the structure of HOXA9 (pdb entry 1PUF) as a search model for HOXB13 and structure of CDX2-DNA$^{meth}$ (pdb entry 5LTY) as a search model for CDX. After the positioning of protein, the density of DNA was clear and the molecule was built manually using COOT (*Emsley et al., 2010*). The rigid body refinement with REFMAC5 was followed by restrain refinement with REFMAC5, as implemented in CCP4

**Table 1.** Data collection and refinement statistics

| | HOXB13-DNA$^{TCG}$ | HOXB13-DNA$^{CAA}$ | CDX2-DNA$^{TCG}$ | CDX2-DNA$^{CAA}$ |
|---|---|---|---|---|
| Data collection | | | | |
| Wavelength (Å) | 0.9724 | 0.9724 | 0.9724 | 0.9724 |
| Resolution range (Å) | 46.29–3.2 (3.31–3.2) | 45.95–2.19 (2.27–2.19) | 43.23–2.57 (2.66–2.57) | 55.96–2.95 (3.13–2.95) |
| Space group | P 2 2 2$_1$ | P 1 2 1 | C 1 2 1 | I 1 2 1 |
| Unit cell (Å, °) | 52.62 52.52 389.33; 90 90 90 | 77.35 57.92 101.28; 90 101.57 90 | 127.95 46.49 68.89; 90 113.27 90 | 70.25 46.69 128.63; 90 101.40 90 |
| Total reflections | 86877 (3476) | 241614 (21747) | 19575 (1958) | 27018 (4003) |
| Unique reflections | 17526 (1361) | 44125 (3912) | 12095 (1197) | 8802 (1264) |
| Multiplicity | 4.2 (3.3) | 5.5 (5.6) | 1.6 (1.6) | 3.2 (3.2) |
| Completeness (%) | 93.0 (90.4) | 97.42 (87.37) | 99.5 (100) | 96.6 (90.5) |
| Mean I/sigma(I) | 7.91 (0.10) | 8.11 (1.10) | 8.47 (2.77) | 7.5 (1.1) |
| R-merge | 0.085 (4.59) | 0.12 (1.21) | 0.13 (5.49) | 0.071 (7.24) |
| R-meas | 0.09 | 0.13 | 0.08 | 0.09 |
| CC1/2 | 0.99 (0.72) | 0.99 (0.71) | 0.99 (0.80) | 0.99 (0.61) |
| Refinement | | | | |
| R-work | 0.21 | 0.25 (0.37) | 0.22 | 0.19 |
| R-free | 0.28 | 0.29 (0.35) | 0.29 | 0.25 |
| Number of non-hydrogen atoms | 5197 | 5591 | 2841 | 2783 |
| macromolecule | 5172 | 5072 | 2748 | 2717 |
| water | 8 | 519 | 93 | 66 |
| Protein residues | 242 | 274 | 144 | 141 |
| RMS (bonds) | 0.025 | 0.011 | 0.018 | 0.012 |
| RMS (angles) | 2.03 | 1.26 | 2.11 | 1.83 |
| Ramachandran favored (%) | 93 | 97 | 97.8 | 99.3 |
| Ramachandran outliers (%) | 1.7 | 0.41 | 1.43 | 0.73 |
| Clashscore | 10.51 | 5.31 | 4.42 | 6.43 |
| Average B-factor | 124.40 | 41.70 | 30.54 | 74.75 |
| macromolecule | 124.70 | 42.10 | 29.30 | 74.41 |

Statistics for the highest-resolution shell are shown in parentheses.
DOI: https://doi.org/10.7554/eLife.32963.015

(*Winn et al., 2011*) and Phenix.refine (*Afonine et al., 2012*). The manual rebuilding of the model was done using COOT. The refinement statistics are presented in *Table 1*. The first seven amino acids from N-termini and the last seven from C-termini were found disordered and were not built in the maps. The end base pairs of the DNA in HOXB13-DNA$^{TCG}$ structure were also found slightly disordered but it was possible to build them to the maps. Figures showing structural representations were prepared using PyMOL (*Schrödinger, 2015*).

## HT-SELEX and motif analysis

MYF6 and BARHL2 HT-SELEX experiments were performed essentially as described in (*Yin et al., 2017*). The PWM models were generated from cycles 3, 3, 4 and 2 of new MYF6 and BARHL2 HT-SELEX reads, and published HT-SELEX reads for HOXB13 (*Yin et al., 2017*) and CDX1 (Methyl-HT-SELEX; *Yin et al., 2017*), respectively, using the multinomial (setting = 1) method (*Jolma et al., 2010*) with the following seeds: HOXB13 single PWM: NCYMRTAAAAN, TCG: NCTCGTAAAAN, CAA: NCCAATAAAAN; CDX1 single PWM: GYMRTAAAA, TCG: GTCGTAAAA, CAA: GCAA-TAAAA; MYF6 single PWM: NRWCAGCTGWYN, AA...TT flank: NAACAGCTGTTN, GT...AC flank: NGTCAGCTGACN; BARHL2 single PWM: NSYTAAWYGNYN, TT: NSYTAATTGNYN, AC: NSYTAAACGKYN.

## Molecular dynamics

Molecular dynamics simulations were performed for HOXB13 complexed with either DNA$^{TCG}$ or DNA$^{CAA}$; the DNA sequence used in the simulations contained nucleotides G5 – C18 from the crystal structure. The CHARMM 36 forcefield (*Best et al., 2012*; *Foloppe and MacKerell, Jr., 2000*; *Hart et al., 2012*; *MacKerell et al., 1998*; *MacKerell et al., 2004*) and CHARMM program (*Brooks et al., 2009*), with the CHARMM interface to OpenMM (*Friedrichs et al., 2009*) to allow the use of NVIDIA graphical processing units (GPUs), were used for all simulations. The starting structure was placed in a cubic solvent box with 8 nm side length containing water (*Jorgensen et al., 1983*) and 0.15 M NaCl; Na$^+$ ions were then added to neutralize the system. After energy minimization to relax initial strain the systems were heated from 100 K to 300 K over 0.1 ns followed by 0.3 ns simulation at constant pressure (1 bar) and constant temperature (300 K), with soft harmonic positional restraints on the protein and DNA atoms. For each complex 3 × 0.8 μs production runs were performed using the GPU, with the pressure and temperature maintained at 1 bar and 300 K, respectively, and without the positional restraints. Particle mesh Ewald summation was used to treat the long range electrostatic interactions, using a sixth order cubic spline interpolation for the charge distribution on the 0.1 nm spaced grid points, kappa = 0.34. The same 0.9 nm cutoff was used for both the direct space part of the PME and for the van der Waals interactions, which were switched to zero from 0.8 nm to 0.9 nm, and the non-bond list was generated with a 1.1 nm cutoff. SHAKE (*Ryckaert et al., 1977*) was used to keep the lengths of all covalent X-H bonds fixed, allowing a time-step of 2 fs.

In the free energy perturbation calculations (*Zwanzig, 1954*) we changed the three base pairs in the TCG sequence into those of the CAA sequence using a total of 43 intermediate states, where the order of change was: turn off charges, change Lennard-Jones parameters, turn on charges. In each state, a 10 ns equilibration was followed by 10 ns production. The free energies were calculated using the Bennett Acceptance Ratio method (*Bennett, 1976*).

## Isothermal titration calorimetry

The ITC experiments were carried similarly to described in Ref. (*Yin et al., 2017*). Briefly, an ITC200 microcalorimeter (MicroCal Inc., Northampton, Massachusetts, USA) in PSF (Protein Science Facility at Karolinska Institute, Sweden) was used to measure binding isotherms of DNAs by direct titration of protein to the cell containing DNA. The measurements were taken at 25°C. Both protein and DNA were prepared in a buffer containing 20 mM HEPES pH 7.5, 300 mM NaCl, 10% glycerol and 2 mM Tris (2-carboxyethyl) phosphine (TCEP). To measure binding affinity, a solution of 0.15 mM protein was titrated to 0.012–0.016 mM solution of DNA. A total of 23 injections were made with 240 s between injections. Each experiment was repeated three times for the reliability of the results. All data were evaluated using the OriginPro 7.0 software package (Microcal) supplied with the calorimeter. The apparent binding constant $K_d$, binding enthalpy $\Delta H$ and stoichiometry n, together with their

corresponding standard deviation (s.d.), were determined by a nonlinear least square fit of the data to standard equations for the binding using a model for one set of independent and identical binding sites as implemented in the package. The entropy and free energy of binding were obtained from the relation $\Delta G = -RT\ln K_d = \Delta H - T\Delta S$.

## Accession codes

The atomic coordinates and diffraction data have been deposited to Protein Data Bank with the accession codes 5EDN and 5EEA, for HOXB13-DNA$^{TCG}$ and HOXB13-DNA$^{CAA}$, respectively and 6ES3 and 6ES2 for CDX2-DNA$^{TCG}$ and CDX2-DNA$^{CAA}$, respectively. All sequence reads are deposited to the European Nucleotide Archive with the study accession number: PRJEB20652

## Acknowledgements

The authors thank Drs. Minna Taipale, Inderpreet Sur, Bernhard Schmierer, Eevi Kaasinen, Sten Linnarsson and Johan Elf for the critical review of the manuscript, Karolinska Institutet Protein Science Facility and Sandra Augsten for protein production, as well as Lijuan Hu and Anna Zetterlund for technical assistance. This work was supported by the Center for Innovative Medicine at Karolinska Institutet, Cancerfonden, the Knut and Alice Wallenberg Foundation and the Swedish Research Council.

## Additional information

### Funding

| Funder | Grant reference number | Author |
|---|---|---|
| Knut och Alice Wallenbergs Stiftelse | KAW 2013.0088 | Jussi Taipale |
| Cancerfonden | CAN 2014/718 | Jussi Taipale |
| Karolinska Institutet | Center for Innovative Medicine | Jussi Taipale |

The funders had no role in study design, data collection and interpretation, or the decision to submit the work for publication.

### Author contributions

Ekaterina Morgunova, Data curation, Formal analysis, Investigation, Visualization, Writing—original draft, Writing—review and editing; Yimeng Yin, Pratyush K Das, Arttu Jolma, Fangjie Zhu, Lennart Nilsson, Data curation, Formal analysis; Alexander Popov, You Xu, Data curation; Jussi Taipale, Conceptualization, Data curation, Supervision, Project administration, Writing—review and editing

### Author ORCIDs

Ekaterina Morgunova http://orcid.org/0000-0002-7754-9021
Pratyush K Das https://orcid.org/0000-0003-2822-9619
Jussi Taipale http://orcid.org/0000-0003-4204-0951

### Decision letter and Author response

Decision letter https://doi.org/10.7554/eLife.32963.031
Author response https://doi.org/10.7554/eLife.32963.032

## Additional files

### Supplementary files

• Supplementary file 1. Thermodynamic characteristics of TF:DNA complexes measured by Isothermal Titration Calorimetry (ITC)*.
DOI: https://doi.org/10.7554/eLife.32963.016

• Transparent reporting form
DOI: https://doi.org/10.7554/eLife.32963.017

## Major datasets

The following datasets were generated:

| Author(s) | Year | Dataset title | Dataset URL | Database, license, and accessibility information |
|---|---|---|---|---|
| Morgunova E, Taipale J | 2016 | Structure of HOXB13-DNATCG | https://www.rcsb.org/structure/5EDN | Publicly available at the RCSB Protein Data Bank (accession no. 5EDN) |
| Morgunova E, Taipale J | 2016 | Structure of HOXB13-DNACAA | https://www.rcsb.org/structure/5EEA | Publicly available at the RCSB Protein Data Bank (accession no. 5EEA) |
| Morgunova E, Taipale J | 2018 | Structure of CDX2-DNATCG | http://www.rcsb.org/structure/6ES3 | Publicly available at the RCSB Protein Data Bank (accession no. 6ES3) |
| Morgunova E, Taipale J | 2018 | Structure of CDX2-DNACAA | http://www.rcsb.org/structure/6ES2 | Publicly available at the RCSB Protein Data Bank (accession no. 6ES2) |
| Yin Y, Taipale J | 2018 | Sequence reads, ENA | https://www.ebi.ac.uk/ena/data/view/PRJEB20652 | Publicly available at the European Nucleotide Archive (accession no. PRJEB20652) |

The following previously published dataset was used:

| Author(s) | Year | Dataset title | Dataset URL | Database, license, and accessibility information |
|---|---|---|---|---|
| Yin Y, Taipale J | 2017 | Impact of cytosine methylation on DNA binding specificities of human transcription factors | https://www.ebi.ac.uk/ena/data/view/PRJEB9797 | Publicly available at the European Nucleotide Archive (accession no. PRJEB9797) |

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
