## [Decision Letter]

Thank you for submitting your article "Two distinct DNA sequences recognized by transcription factors represent enthalpy and entropy optima" for consideration by *eLife*. Your article has been reviewed by three peer reviewers, and the evaluation has been overseen by a Reviewing Editor and Naama Barkai as the Senior Editor. The following individuals involved in review of your submission have agreed to reveal their identity: Gary Stormo (Reviewer #1); Thomas D Tullius (Reviewer #3).

The reviewers have discussed the reviews with one another and the Reviewing Editor has drafted this decision to help you prepare a revised submission.

Summary:

In this manuscript, the authors address an important but poorly understood phenomenon: that DNA-binding proteins sometimes recognize with similar affinity DNA sites that diverge substantially in sequence. For two homeodomain proteins, they assemble thermodynamic data that argue for the primary role of entropy-enthalpy compensation in producing similar binding affinity for distinct sequences. They also collect thermodynamic binding data for two additional proteins that exhibit the same behavior, supporting their conclusions. They interpret structural data (X-ray co-crystal structures) as supporting the idea that ordered water structure (or its lack) at the protein-DNA interface is the main source of this thermodynamic compensation.

Previous work in this area (particularly the important paper by Jen-Jacobson et al. that is referenced by the authors) focused on thermodynamic analysis of several different DNA binding proteins bound to their canonical binding sites, and noted the entropy-enthalpy compensation phenomenon. The present manuscript instead shows that entropy-enthalpy compensation occurs for the same protein bound to two very different DNA sites, a new and significant observation. (Notably, in the data presented in Figure 1 of the Jen-Jacobson et al. paper, entropy-enthalpy compensation is apparent for proteins binding to the same core site in different sequence contexts, and also for proteins binding to point mutants of the canonical site, similar in some ways to the observations central to the current manuscript. However, Jen-Jacobson et al. do not discuss this point.) The authors of the present manuscript significantly extend previous work in this area by solving X-ray co-crystal structures of the complexes of interest, and by analyzing these structures to provide a structural underpinning for the thermodynamic observations.

Opinion:

The manuscript will be influential in understanding the important problem of how proteins select DNA binding sites in genomes.

Essential revisions:

A) Conceptual:

1) The generalizability of the finding based on a handful of examples of TFs binding to DNA, to all other biomolecular interactions, is probably somewhat overstated. For example, how sure are we at this point that the last sentence of the Abstract will turn out to be as generalizable as it is currently stated? The authors should provide stronger arguments or more evidence that such duality of enthalpy/entropy-driven binding actually exists among many interactions. Or tune down this and related statements.

2) In a general sense, it would be more reassuring to see correlations of how this mechanism might translate to in vivo TF functions.

3) One of the surprising results is that, as stated "no substitutions were identified that would lead to a specific loss of binding to the TCG or CAA sequences". This observation should be better explained.

4) When discussing the presence or absence of structured water in the complexes (a key feature of their structural analysis), the authors state: "A relatively large solvent channel (6.4 Å in smallest diameter) exists between the α3 helix of HOXB13 and DNA (Figure 4D)." To clarify their argument, the authors should specify in the text that this observation refers to the HOXB13-DNA-TCG structure (as is done in the figure caption). Is such a solvent channel present in the HOXB13-DNA-CAA structure? The text seems to imply that it is not present in the CAA structure, but please clarify. Similarly, there is no mention in the text as to the presence or absence of a channel in the CDX2 complexes. A central point in the authors' connection of structure with thermodynamics for these systems is the presence or absence of ordered water at the protein-DNA interface. Differences in the channel are implied to be, at least in part, the reason for different water ordering that is reflected in the entropy changes associated with binding, so it is important to clarify whether this is indeed observed in the structures.

B) Presentation:

5) The result, if generalizable, is clearly significant, although the authors could more explicitly spell out what findings are truly novel versus already established. For example, it is already established that binding of a TF to very different sequences is relatively widespread, but it is less well established how the thermodynamic underpinnings of such differential recognition mechanisms operate.

6) It is not immediately clear how new the observations presented in Figure 1 are. How much of what is presented here is based on data obtained in previous papers?

7) The flow might be quite difficult to follow for those without direct expertise in the technical details of the approaches used. Here is a suggestion for reorganization to improve the flow: The primary (and most obvious) evidence for the mechanistic difference in binding for the two homeodomains with the two DNA sequences is the calorimetry data shown in Figure 5. The X-ray structural data are interesting, but not really conclusive as to mechanism; the structures support the thermodynamic data but don't really stand alone. For example, that all four structures look like every other homeodomain-DNA structure is not surprising at all. The differences between the four structures are pretty subtle, and also are not well-explained in the manuscript for the two CDX2 complexes. In contrast, the calorimetry data are very consistent for the two homeodomains, and for two other proteins. Thus, we suggest to reorder the Results section as follows: (1) PWM models (which suggest the problem to be addressed); (2) calorimetry data (to experimentally address mechanism); (3) X-ray data (to support the calorimetry data); (4) MD simulations (that support the X-ray and calorimetry).

8) The references in the main text are all over the place, and should be reformatted according to *eLife* referencing style.

---

## [Author Response]

Essential revisions:A) Conceptual:1) The generalizability of the finding based on a handful of examples of TFs binding to DNA, to all other biomolecular interactions, is probably somewhat overstated. For example, how sure are we at this point that the last sentence of the Abstract will turn out to be as generalizable as it is currently stated? The authors should provide stronger arguments or more evidence that such duality of enthalpy/entropy-driven binding actually exists among many interactions. Or tune down this and related statements.

We agree that the arguments made in the original manuscript did not clearly distinguish between theory and observations. We have now clarified that the phenomenon we observe exists due to the fact that entropy and enthalpy of binding are partially independent variables. They are often anticorrelated, but not completely. If they were completely anticorrelated, all binding affinities would be identical, which is clearly not the case. We have now clarified this in the third and fourth paragraphs of the subsection “Conclusions”. We have also clarified that even when the phenomenon itself is general, its relative importance in particular interactions varies. Because we selected cases with two equally strong maxima, the effect is stronger here than in most interactions. The relative importance of the phenomenon across other types of interactions needs to be evaluated experimentally. This is now stated in the fourth paragraph of the aforementioned subsection.

2) In a general sense, it would be more reassuring to see correlations of how this mechanism might translate to in vivo TF functions.

We agree, and have now added a citation to Yin et al., 2017 and Zuo et al., 2017 papers, which show that one of the HOXB13 optima, TCG, is affected by DNA methylation (Conclusions, second paragraph). Methylation of this site further increases HOXB13 affinity. The CAA site does not contain CpG dinucleotide, and is not epigenetically methylated. Thus, the ability of HOXB13 to bind to two distinct sites with similar affinity allows evolution of two types of regulatory sequences, those that are directly and positively affected by methylation, and those that are not.

3) One of the surprising results is that, as stated "no substitutions were identified that would lead to a specific loss of binding to the TCG or CAA sequences". This observation should be better explained.

We have now clarified that this set of experiments was carried out to mechanistically understand the origin of the dual specificity, and the molecular mechanisms that explain different specificities between the posterior HOX proteins. This analysis failed to find mutations that lead to selective and complete loss of binding to TCG or CAA site. Some weaker effects were detected, and this is now clarified in the fifth paragraph of the subsection “Structural analysis of HOXB13 and CDX2 bound to DNA^TCG^ and DNA^CAA^”. As we did not test all possible mutations, we cannot conclude that no such mutations exist. However, the failure to find highly selective mutations is consistent with the thermodynamic model presented, as both TCG and CAA are bound by the same conformation of the HOXB13 protein, using the same contacting amino-acids (even when the contacts can occur via water or be direct). This is now stated in the first paragraph of the subsection “Conclusions”.

4) When discussing the presence or absence of structured water in the complexes (a key feature of their structural analysis), the authors state: "A relatively large solvent channel (6.4 Å in smallest diameter) exists between the α3 helix of HOXB13 and DNA (Figure 4D)." To clarify their argument, the authors should specify in the text that this observation refers to the HOXB13-DNA-TCG structure (as is done in the figure caption). Is such a solvent channel present in the HOXB13-DNA-CAA structure? The text seems to imply that it is not present in the CAA structure, but please clarify. Similarly, there is no mention in the text as to the presence or absence of a channel in the CDX2 complexes. A central point in the authors' connection of structure with thermodynamics for these systems is the presence or absence of ordered water at the protein-DNA interface. Differences in the channel are implied to be, at least in part, the reason for different water ordering that is reflected in the entropy changes associated with binding, so it is important to clarify whether this is indeed observed in the structures.

We thank the reviewers for pointing this out. We have now clarified that the water channel exists in both CDX and HOXB13 TCG structures, but not in the corresponding CAA structures (subsection “Role of water molecules in the protein-DNA interface”). We have also added new figure supplement to Figure 4 to illustrate this.

B) Presentation:5) The result, if generalizable, is clearly significant, although the authors could more explicitly spell out what findings are truly novel versus already established. For example, it is already established that binding of a TF to very different sequences is relatively widespread, but it is less well established how the thermodynamic underpinnings of such differential recognition mechanisms operate.

We have now clarified in the Introduction (third paragraph) and in legend to Figure 1 that the ability of TFs to bind to two distinct sequences is well established.

In addition, we have clarified that the mechanism for this effect is also understood for dimers (e.g. SREBP; Introduction, third and fourth paragraph), but not for some monomeric TFs.

6) It is not immediately clear how new the observations presented in Figure 1 are. How much of what is presented here is based on data obtained in previous papers?

We have clarified that Figure 1 shows new motifs derived from existing data (panels C, D) or from new data (panel A, B). We have now also cited several publications, including Badis et al., 2009, Jolma et al., 2013 and Zuo et al. 2017 to clarify that two distinct binding models for HOXB13 have been previously reported (subsection “Modeling the binding of many TFs requires more than one PWM model”, first paragraph).

7) The flow might be quite difficult to follow for those without direct expertise in the technical details of the approaches used. Here is a suggestion for reorganization to improve the flow: The primary (and most obvious) evidence for the mechanistic difference in binding for the two homeodomains with the two DNA sequences is the calorimetry data shown in Figure 5. The X-ray structural data are interesting, but not really conclusive as to mechanism; the structures support the thermodynamic data but don't really stand alone. For example, that all four structures look like every other homeodomain-DNA structure is not surprising at all. The differences between the four structures are pretty subtle, and also are not well-explained in the manuscript for the two CDX2 complexes. In contrast, the calorimetry data are very consistent for the two homeodomains, and for two other proteins. Thus, we suggest to reorder the Results section as follows: (1) PWM models (which suggest the problem to be addressed); (2) calorimetry data (to experimentally address mechanism); (3) X-ray data (to support the calorimetry data); (4) MD simulations (that support the X-ray and calorimetry).

We agree that a different flow of the manuscript might make it easier for some scientists to follow the manuscript. However, we prefer to retain the order of presentation as it is collinear with the actual process of the study, and the origin of the hypothesis that was tested. This is in principle an important aspect of the scientific process, and we feel that giving the reader a sense of how the discovery was made is more important than the relatively minor increase in clarity that the reordering might accomplish.

8) The references in the main text are all over the place, and should be reformatted according to eLife referencing style.

Thank you for pointing this out. We have now reformatted the references according to the house style.